# NMR Profiling of *Ononis diffusa* Identifies Cytotoxic Compounds against Cetuximab-Resistant Colon Cancer Cell Lines

**DOI:** 10.3390/molecules26113266

**Published:** 2021-05-28

**Authors:** Vittoria Graziani, Nicoletta Potenza, Brigida D’Abrosca, Teresa Troiani, Stefania Napolitano, Antonio Fiorentino, Monica Scognamiglio

**Affiliations:** 1Dipartimento di Scienze e Tecnologie Ambientali, Biologiche e Farmaceutiche, Università degli Studi della Campania “Luigi Vanvitelli”, Via Vivaldi 43, 81100 Caserta, Italy; v.graziani@qmul.ac.uk (V.G.); nicoletta.potenza@unicampania.it (N.P.); brigida.dabrosca@unicampania.it (B.D.); 2Oncologia medica, Dipartimento di Medicina di precisione, Università degli Studi della Campania “Luigi Vanvitelli”, S. Andrea delle Dame, Via L. De Crecchio 7, 80138 Napoli, Italy; teresa.troiani@unicampania.it (T.T.); stefania.napolitano@unicampania.it (S.N.)

**Keywords:** cytotoxic activity, natural products, NMR, *Ononis diffusa*, *Ononis variegata*, oxylipins

## Abstract

In the search of new natural products to be explored as possible anticancer drugs, two plant species, namely *Ononis diffusa* and *Ononis variegata*, were screened against colorectal cancer cell lines. The cytotoxic activity of the crude extracts was tested on a panel of colon cancer cell models including cetuximab-sensitive (Caco-2, GEO, SW48), intrinsic (HT-29 and HCT-116), and acquired (GEO-CR, SW48-CR) cetuximab-resistant cell lines. *Ononis diffusa* showed remarkable cytotoxic activity, especially on the cetuximab-resistant cell lines. The active extract composition was determined by NMR analysis. Given its complexity, a partial purification was then carried out. The fractions obtained were again tested for their biological activity and their metabolite content was determined by 1D and 2D NMR analysis. The study led to the identification of a fraction enriched in oxylipins that showed a 92% growth inhibition of the HT-29 cell line at a concentration of 50 µg/mL.

## 1. Introduction

Natural products play a crucial role in the discovery of anticancer compounds even today, when new technologies are available to easily obtain a wide range of drug candidates. The success of natural products is inherent to their structure: they possess a well-defined three-dimensional scaffold with functional groups precisely oriented in space. Furthermore, they are characterized by an enormous structural and functional diversity [1,2]. These features confer them with extraordinary selectivity and specificity, when compared to artificially designed molecules [3]. Plants, in particular, are a unique source of chemicals since they use specialized metabolites to interact with the environment and in response to several biotic and abiotic stresses [4,5]. Humankind has, therefore, always exploited this ability to produce a wide and diversified array of chemicals [6].

In recent years, the need for anticancer compounds has become particularly pressing and, in this context, colorectal cancer is definitely under the spotlight. This cancer is indeed one of the leading causes of cancer-related death worldwide and one of the most frequently diagnosed malignant diseases in Europe [7]. Although the outcomes of patients with metastatic colorectal cancer (mCRC) has improved in recent years [8], new challenges are on the horizon. Nowadays, resistance to both chemotherapy and molecularly targeted therapies represents a major problem for setting up effective treatments. Specifically, clinical data highlight the emergence of acquired resistance to anti-EGFR therapies [9]. EGFR (Epithelial Growth Factor Receptor) is a transmembrane tyrosine kinase receptor that, once activated, triggers two main signaling pathways that are involved in cell proliferation, survival and motility [10]. Drugs like the monoclonal antibody cetuximab are effective at blocking the EGFR receptor. However, patients carrying mutations at some of the intracellular effectors of EGFR activation have intrinsic resistance to cetuximab treatment [11]. Furthermore, around 25% of cetuximab-sensitive mCRC patients develop, after an initial response, secondary resistance to this drug [11,12]. Therefore, it is urgent to find chemotherapeutic agents to prevent or overcome the limit imposed by these resistances to the effectiveness of the anticancer drug.

In the search for compounds with anticancer activity in plants of Mediterranean region, we recently screened a set of plants of the Fabaceae family with a metabolomics-based approach [13]. This study led to the identification of candidate molecules that are currently under further investigation. In fact, there appears to be a link between members of the Fabaceae family and colon cancer prevention and therapy [13].

Herewith, we undertook the study of two further Fabaceae plants which were not analyzed in our previous study: *Ononis diffusa* Ten. and *Ononis variegata* L. *Ononis* is a large genus of perennial herbs and shrubs, including food, foraging and medicinal plants [14]. Several *Ononis* species have been studied, and bioactive compounds including flavonoids, isoflavonoids, alkylresorcinols, coumarin derivatives, and terpenes have been reported from them [14,15,16,17]. However, no phytochemical studies have been previously carried out on *O. diffusa* and *O. variegata*.

Extracts from the two species which are the objects of this study were first screened for their antiproliferative activity on colorectal cancer cell lines (Caco-2, HT-29, and HCT-116) characterized by different genetic profiles. Specifically, the Caco-2 cell line is sensitive to cetuximab as it has no genetic defects associated with anti-EGFR therapy resistance. Meanwhile, the HT-29 and HCT-116 cell lines are intrinsically resistant to cetuximab since they harbor BRAF V600E and KRAS/PIK3CA mutations, respectively [18]. *Ononis diffusa* showed a marked cytotoxic effect in all the cell lines and especially in those exhibiting drug resistance. Thus, *O. diffusa* crude extract was further tested on cetuximab-sensitive (GEO and SW48) and on acquired cetuximab-resistant (GEO-CR and SW48-CR) human colon cancer cell lines. GEO-CR and SW48-CR represent a preclinical model for the study of acquired resistance to cetuximab. The potential cytotoxicity of the extracts was evaluated through MTT assays. In the attempt to identify the metabolites responsible for the activity, NMR profiles of the extracts were obtained. Furthermore, a partial purification, always paired with the NMR analysis, and biological tests led to the identification of a mixture of oxylipins as putative bioactive compounds responsible for the antiproliferative properties of the crude extract.

## 2. Results

### 2.1. Cytotoxic Activity

#### 2.1.1. Cytotoxic Activity of Crude Extracts

Plant extracts were obtained with a mixture of methanol and water (1:1). The potential antiproliferative activity of the plant crude extracts was evaluated on a panel of colon cancer cell lines (Caco-2, HT-29 and HCT-116). While *O. variegata* showed no inhibition of cell proliferation at any of the tested concentrations, *O. diffusa* showed a significant inhibition against all of the cell lines, beginning at a concentration of 50 µg/mL (Figure 1). *Ononis diffusa* extract exhibited a more pronounced effect on HT-29 cell line.

#### 2.1.2. Cytotoxic Activity of the Crude Extract against Colon Cancer Cell Lines with Acquired Resistance to Cetuximab

Based on the significant activity shown by *O. diffusa* extract on the three colon cancer cell lines, including the intrinsically cetuximab-resistant ones, this extract was further explored for its activity against both cetuximab-sensitive (GEO and SW48) and secondary cetuximab-resistant cancer cell lines (GEO-CR and SW48-CR). The extract showed a remarkable inhibition of the proliferation of the acquired resistant cell lines (Figure 2). Indeed, while a growth inhibition of 50% on SW48 cells was observed at the highest tested concentration (250 µg/mL), a 50% inhibition of the corresponding secondary resistant cell line was observed at a dose as low as 50 µg/mL.

On the other hand, the activity on the GEO and GEO-CR cell lines was comparable; important effects were observed starting from a concentration of 150 µg/mL of the extract.

### 2.2. NMR Profiling of the Extracts

NMR spectra were obtained for the extracts of the two Ononis species. The plant material was extracted with a mixture of methanol-*d*_4_ and phosphate buffer in D_2_O (1:1) and the solution thereby obtained was analyzed by NMR.

Although *O. variegata* showed no activity in the biological tests, a comparison of the profiles of the two species helped us narrow down the set of the possible bioactive compounds. It was clear, indeed, from the comparison of the ^1^H-NMR spectra (Figure 3), that there were metabolites present in both species that, furthermore, were the main components of *O. variegata*. These metabolites could not be considered responsible for the activity observed for *O. diffusa* extract. Besides several sugars, amino acids, and organic acids identified on the basis of literature data [19,20], *O. variegata* also showed signals of caffeic acid and of trigonelline (Figure 3) [20,21].

All these metabolites were also detected in the *O. diffusa* extract. Other signals only present in *O. variegata* were very likely attributable to caffeoylquinic acids. However, as these signals were not detected in the active extract, the identification of the metabolites generating them was beyond the scope of the present work.

The aromatic and olefinic region of *O. diffusa* ^1^H-NMR spectrum showed several peaks that were not detected in *O. variegata* (Figure 3). Further differences were observed in the aliphatic region; protons at δ_H_ 1.23 and a triplet at δ_H_ 0.83 indicated the presence of alkyl chains. These signals are usually attributed to fatty acids. However, the solvent mixture herewith used would not explain the extraction of these compounds, which seem to be rather abundant in the extract.

In the attempt to assign signals to metabolites only present in *O. diffusa*, and therefore very likely responsible for the activity, an extensive 2D-NMR study of the extract was carried out. However, the main correlations detected for the signals in the aromatic region did not allow us to further identify the unknown metabolites. While the HSQC and COSY (Appendix A) experiments suggested the presence of further 1,2,4-trisubstituted ring systems, the long-range correlation experiments were not determining, since the main correlations detected were those of the already-known compounds (whose identity was therefore confirmed, Appendix A). Concerning the signals in the aliphatic region, the presence of an alkyl chain was undoubtful, based on the NMR data previously discussed. However, long-range correlations of protons resonating in this region with carbon signals in the range of 60–80 ppm suggested the possibility that the fatty acids present in the extracts could be oxygenated, as in the case of oxylipins.

### 2.3. Partial Purification of the Extract and Biological Activity

Since the 2D NMR of *O. diffusa* extract was dominated by the signals of the already-known compounds, a partial purification of a larger quantity of extract was carried out. The extract was first partitioned with water and ethyl acetate. The water fraction obtained was chromatographed on amberlite (XAD-4 and XAD-7). The columns were eluted with methanol first, and then with water. The methanolic fractions were joined together due to their very similar TLC profiles. The fraction (OdM) thereby obtained was analyzed by NMR and the ^1^H-NMR spectrum, along with the HSCQ experiment, clearly showed that the aim was achieved (Figure 4). The OdM spectra did not show signals of primary metabolites, caffeic acid, or trigonelline (eluting in the water fraction).

However, the spectrum was still rich in signals, especially in the aromatic region. The HSQC experiment (Figure 4) and the other 2D NMR experiments revealed several correlations. In particular, it was observed that the singlet proton at δ_H_ 8.14 was not bound to a carbon atom; this value was in good agreement with a hydrogen atom of an amidic functional group. This signal heterocorrelated, in the CIGAR-HMBC experiment (Appendix A), with a carbonyl carbon at δ_C_ 176.5. Furthermore, it also correlated, in the same experiment, with the quaternary carbon at δ_C_ 124.2, which was in turn correlated with two protons of a 1,2,4-tribubstituted aromatic system resonating at δ_H_ 6.90 (dd, *J* = 8.2; 2.0 Hz) and 6.97 (d, *J *= 2.0 Hz). The latter signal showed cross peaks with two oxygenated aromatic carbons at δ_C_ 147.8 and 146.1, both of which correlated with a third aromatic proton at δ_H_ 6.76 (d, *J *= 8.2 Hz) and with a singlet methylene at δ_H_ 5.87 (δ_C_ 101.2), indicating the presence of a 3,4-dioxymethylenephenyl group bound to the amide nitrogen. In addition, in the CIGAR experiment, the amidic proton correlated with a further aromatic carbon at δ_C_ 157.9, which in turn displayed cross-peaks with a proton at δ_H_ 8.01 (d, *J *= 9.0 Hz), as an indicator of a second 1,2,4-tribubstituted aromatic system. This proton correlated in the long-range experiment with the amide carbonyl and with a quaternary aromatic carbon at δ_C_ 118.8, which in turn correlated with other two aromatic protons resonating at δ_H_ 7.07 (d, *J *= 2.4 Hz) and 7.22 (dd, *J *= 9.0; 2.4 Hz), showing cross-peaks with an oxygenated aromatic carbon at δ_C_ 162.2. This carbon correlated with an anomeric proton at δ_H_ 4.98, therefore suggesting the presence of a sugar moiety, putatively identified, on the basis of an HSQC-TOCSY experiment, as glucose. These data were in agreement with the presence of a glucopyranosyl-2-hydroxy-*N*-(3,4-dioxymethylenephenyl)benzamide. However, the isolation and complete structural elucidation of this compound will be needed to confirm the hypothesized structure.

More signals were detected in the aromatic region. Signals belonging to the A-ring of flavonoids were observed as two COSY-correlating doublets (*J *= 2.0 Hz) at δ_H_ 6.20 and 6.39. These protons were correlated in the HSQC experiment to the carbons at δ_C_ 93.1 and 98.4, respectively. Furthermore, the former proton also showed long-range correlations with the two carbons resonating at δ_C_ 161.2 and 165.0. This second carbon was also correlated to the δ_H_ 6.39, which showed further long-range correlations with the carbons at δ_C_ 156.9 and 104.0. It was not possible to unambiguously identify these metabolites, due to the lack of further correlations, but the NMR data here described prompted us to hypothesize that the flavonoids were all characterized by a hydroxy function bound to the C-3 [21].

Finally, the signals in the range of 6.8–7.4 ppm were correlated, in the HSQC experiment, with the carbons in the range of 125–145 ppm, suggesting the presence of olefinic protons. These protons also showed COSY correlations with protons resonating in the range of 5.9–6.1 ppm bound to carbons in the range of 120–125 ppm (Appendix A). The proton signals at 5.9–6.1 ppm showed further COSY correlations with signals in the aliphatic region (2.4–2.8 ppm), in turn correlating with protons geminal to oxygen. Since the mixture was complex and these compounds were minor components, it was not possible to definitely identify the compounds. However, it is plausible that these signals belonged to oxylipins.

The OdM fraction was tested on HT-29 cells (Figure 5), but no activity was observed. Two possible explanations were then taken into consideration: (i) the occurrence of synergistic effects in the crude extract or (ii) the fact that the OdM fraction obtained from the amberlite chromatography, although enriched in several specialized metabolites, did not contain a significant amount of the active compounds. These active metabolites could be then found either in the water fraction obtained from the same chromatography, or in the ethyl acetate fraction (OdE) obtained in the previous step. By comparing the NMR profiles of the OdM and the crude extract (Figure 3 and Figure 4), it was clear that, besides the signals belonging to metabolites also detected in the crude extract of the inactive species *O. variegata*, there was also a significant decrease in the intensity of signals so far putatively attributed to oxylipins. These oxylipins could have been in the OdE fraction, which was therefore assayed. The ethyl acetate fraction showed activity on HT-29 cells (Figure 5).

The inhibition was of 92% at a concentration of 50 µg/mL, and around 40% at the concentration of 10 µg/mL. The ^1^H-NMR of the ethyl acetate extract confirmed that this fraction was enriched in signals that were only detected in traces in the methanol fraction, and that could be attributed to oxylipins (Figure 6).

Besides the triplet at δ_H_ 0.83 and the methylene signals at δ_H_ 1.23, signals belonging to two olefinic protons conjugated to a carboxylic function were detected at δ_H_ 5.90 and δ_H_ 7.00. Allylic protons were instead resonating in the range 2–3 ppm, while signals in the region between 3 and 5 ppm were attributable to protons geminal to hydroxyl groups. The hypothesized structure is reported in Figure 6. This hypothesis was drawn based on the analysis of the 2D-NMR spectra of the extract previously discussed.

A complete structural elucidation of these metabolites was not possible in mixture, because it is very likely to have been made up of compounds with variable chain lengths. However, it is also clear that compounds with different substitution patterns were present in the mixture, including compounds that might bear acetyl groups (as shown by singlets at around 2 ppm).

## 3. Discussion

The present study demonstrates that *O. diffusa* is a source of potential anticancer compounds acting on drug-resistant colon cancer cell lines. The aim of the present work was not only to identify cytotoxic extracts, but particularly sources of compounds able to overcome anti-EGFR therapy resistance in mCRC.

EGFR is a very important target in cancer therapy [11], being a central regulator of tumor progression in a variety of human cancers, including mCRC [22], one of the leading causes of cancer-related death worldwide. One way to inhibit the activation of EGFR is with monoclonal antibodies like cetuximab [23]. Unfortunately, primary resistance (due to specific mutations) to cetuximab in mCRC patients, who therefore do not respond to this treatment, has been reported [24]. Furthermore, it has been shown that one fourth of cetuximab-sensitive mCRC patients develop secondary resistance [11,12]. The emergence of secondary resistance might be due to the selection of drug-insensitive subclones imposed by the continuous EGFR blockade [24]. Finding drugs acting with alternative modes of action, able to overcome or bypass these innate and acquired resistances is therefore crucial. In this study, we used Caco-2, GEO, and SW48 cells as cetuximab-sensitive models, HT-29 and HCT-116 as intrinsically cetuximab-resistant models and GEO-CR and SW48-CR as cell models with acquired resistance to cetuximab. Although the data here reported are preliminary and extensive tests on pure compounds are needed to assess their activity, toxicity, and modes of action, the extract herewith analyzed showed a strong inhibition of cell growth on all the cell lines. Of note, the activity on cetuximab-resistant human cancer cell lines was remarkable. In particular, the extract strongly inhibited the growth of HT-29, a cell line harboring a BRAF mutation, which is a strong negative prognostic biomarker for patients suffering from mCRC [12].

Based on the partial purification and on the fraction testing and profiling, we could identify the class of compounds potentially responsible for the activity exerted by the extracts. It was therefore suggested that the oxylipin components of the extract could be the compounds responsible for the biological activity. The crude extract demonstrated strong cytotoxicity (Figure 1 and Figure 2). Based on a comparison to the inactive extract of the related plant *O. variegata* (Figure 3), which was also tested and analyzed, it was possible to exclude some of the metabolites from the list of candidate bioactive compounds. These compounds (caffeic acid, caffeoyl derivatives, trigonelline, and several primary metabolites) were, indeed, the main components of the *O. variegata* extract (Figure 3), which however showed no activity even at the highest tested concentration. A second level of selection was obtained after the partial purification of the crude extract: OdE and OdM fractions were obtained when the extract was partitioned with ethyl acetate/water and then the aqueous fraction therefrom was further purified on amberlite. Although the OdM fraction was particularly enriched in phenolic compounds, it did not show activity against HT-29 cells (Figure 4 and Figure 5). On the other hand, the OdE fraction was very active and even more strongly inhibiting of HT-29 cell growth than the crude extract. Based on the NMR analysis, it was possible to tentatively identify the main compounds in this fraction as oxylipins (Figure 6 and Appendix A).

Oxylipins are oxidized fatty acids, used by plants mostly as signaling molecules [25,26], to the best of our knowledge. Jasmonic acid is by far the most studied oxylipin, due to its central role as a plant hormone involved in the regulation of developmental and defense-related processes [27]. Given these roles in signaling and as plant hormones, oxylipins are usually present at low concentrations [25] and therefore have also been seldom studied for further biological activities. It has been suggested that these molecules promote apoptosis in animal cells by altering the intracellular calcium signaling and inducing cytoskeletal instability [28], although the molecular mechanism is not yet known. Besides jasmonates, different substitute oxylipins have been reported, including oxylipins with cytotoxic, anti-inflammatory, and potential anticancer properties [29,30]. The oxylipins in *O. diffusa* extract seemed to be polyoxygenated compounds. This hypothesis was supported by their polarity and chromatographic behavior.

Although the oxylipins were the most abundant compounds in the active fraction, we cannot completely exclude the possibility that less-abundant compounds could have been responsible for the observed activity. However, a further effort to perform a complete isolation and structural elucidation of the pure compounds will be carried out as the object of future studies. The observed biological activity makes the OdE fraction a potential source of compounds that could be further explored with the aim of finding drug candidates able to overcome either intrinsic or acquired drug resistance in colorectal cancer cells.

## 4. Materials and Methods

### 4.1. Plant Sampling

Plant samples belonging to the two species, *O. diffusa* and *O. variegata* were harvested in the spring at the “Castel Volturno” Nature Reserve (40°57.587′ N, 14°00.105′ E; southern Italy) and identified by Prof. Assunta Esposito. Voucher specimens were deposited at the Herbarium of DiSTABiF of Università degli Studi della Campania “Luigi Vanvitelli.” Leaf samples were collected and immediately frozen in liquid N_2_ to avoid unwanted enzymatic reactions, and stored at −80 °C before the freeze-drying process. Lyophilized samples were powdered in liquid nitrogen and stored at −20 °C.

### 4.2. Plant Chemical Profile

#### 4.2.1. Extraction

An aliquot (50 mg) of freeze-dried and powdered plant material extracted with 1.5 mL of phosphate buffer (Fluka Chemika, Buchs, Switzerland; 90 mM; pH 6.0) in D_2_O (Cambridge Isotope Laboratories, Andover, MA, USA) containing 0.1% *w/w* trimethylsilylpropionic-2,2,3,3-*d_4_* acid sodium salt (TMSP, Sigma-Aldrich, St. Louis, MO, USA) and CD_3_OD (Sigma-Aldrich, St. Louis, MO, USA) (1:1). The mixture was vortexed at room temperature for 1 min, ultrasonicated (Elma Transsonic Digital, Hohentwiel, Germany) for 40 min, and centrifuged (Beckman Allegra™ 64R, F2402H rotor; Beckman Coulter, Fullerton, CA, USA) at 13,000 rpm for 10 min. A volume of 0.65 mL was transferred to a 5 mm NMR tube and analyzed by NMR [31].

#### 4.2.2. NMR Analysis

NMR spectra were recorded at 25 °C on a Varian Mercury Plus 300 Fourier transform NMR operating at 300.03 MHz for ^1^H and 75.45 MHz for ^13^C. CD_3_OD was used as the internal lock. 1D and 2D NMR spectra were acquired using Varian standard pulse sequences and as previously described [13].

### 4.3. Partial Purification of O. diffusa Extract

Dried leaves (60 g) of *O. diffusa* were powdered and underwent three cycles of an ultrasound-assisted extraction with a MeOH:H_2_O (1:1) solution (1.8 L), obtaining a crude extract (16 g), which was dissolved in H_2_O and separated by liquid–liquid extraction using EtOAc as an extracting solvent, obtaining an ethyl acetate fraction (OdE; 3 g) and a water fraction. The aqueous fraction was chromatographed on Amberlite XAD-4 and XAD-7 with water and then with methanol. The two alcoholic eluates were joined together to give the OdM fraction (1.7 g). Aliquots were used for bioassays and NMR analyses.

### 4.4. Cell Lines

#### 4.4.1. Cell Cultures

Human HCT-116, HT-29, Caco-2, and SW48 colorectal cancer cell lines were obtained from the American Type Culture Collection (ATCC) (Manassas, VA, USA). The human GEO colon cancer cell was a gift of Dr. N. Normanno (National Cancer Institute, Naples, Italy). The HCT-116 and HT-29 cancer cells were cultured in RPMI 1640 medium (Lonza, Cologne, Germany), and supplemented with 10% FBS, 2 mM L-glutamine, 50 U/mL penicillin, and 100 µg/mL streptomycin (Lonza, Cologne, Germany). Caco-2 cell line was cultured in DMEM medium (Lonza, Cologne, Germany), and supplemented with 10% FBS, 2 mM L-glutamine, 1% non-essential amino acid, 50 U/mL penicillin, and 100 µg/mL streptomycin (Lonza, Cologne, Germany). The GEO and GEO-CR clones were grown on DMEM medium supplemented with 20% FBS, 1% penicillin/streptomycin (Lonza, Cologne, Germany). The SW48 and SW48-CR cells were cultured in RPMI 1640 medium supplemented with 10% FBS and 1% penicillin/streptomycin. All cell lines were maintained in a humified atmosphere of 95% air and 5% CO_2_ at 37 °C, and routinely screened for the presence of mycoplasma (Mycoplasma Detection Kit, Roche Diagnostics, Basel, Switzerland).

#### 4.4.2. Proliferation Assay

Cell proliferation was measured with 3-(4,5-dimethylthiazol-2-yl)-2,5-diphenyltetrazolium bromide (MTT) assay [32]. Briefly, cells in the logarithmic growth phase were plated in 96-well plates and incubated for 24 h before exposure to increasing doses of plant extracts (10, 50, 100, 150, 200 and 250 µg/mL). At 48 h after treatment, 50 µL of 1 mg/mL (MTT) were mixed with 200 μL of medium and added to the well. At 1 h after incubation at 37 °C, the medium was removed, and the purple formazan crystals produced in the viable cells were solubilized in 100 μL of dimethyl sulfoxide and quantitated by measurement of absorbance at 570 nm with a plate reader. Results were reported as mean +SD of % of cell growth respect to the control, from six replicates. The control was represented by 0.25% DMSO treatment, corresponding to the higher amount of DMSO used for the tests.

#### 4.4.3. Statistical Analyses

Bioassays were carried out in six replicates. Statistical analyses were performed using Excel 2010 (Microsoft Corporation; Redmond, WA, USA). A Student’s *t* test (*p* < 0.001) was used to determine the statistical significance of the experimental results.

## Figures and Tables

**Figure 1 molecules-26-03266-f001:**
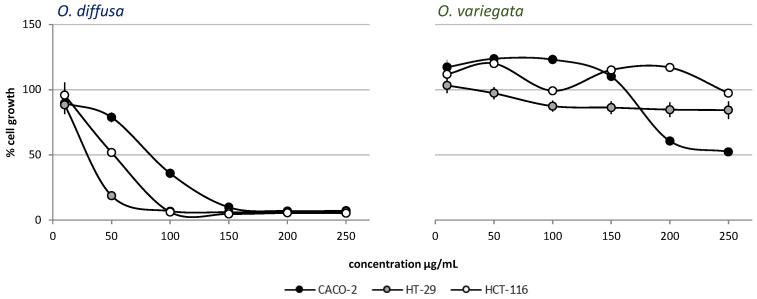
Antiproliferative activity evaluation of *O. diffusa* and *O. variegata* on the Caco-2, HT-29, and HCT-116 cell lines. Cell growth is expressed as percentage from control, and is plotted on the vertical axis, while doses of plant extracts are reported on the horizontal axis.

**Figure 2 molecules-26-03266-f002:**
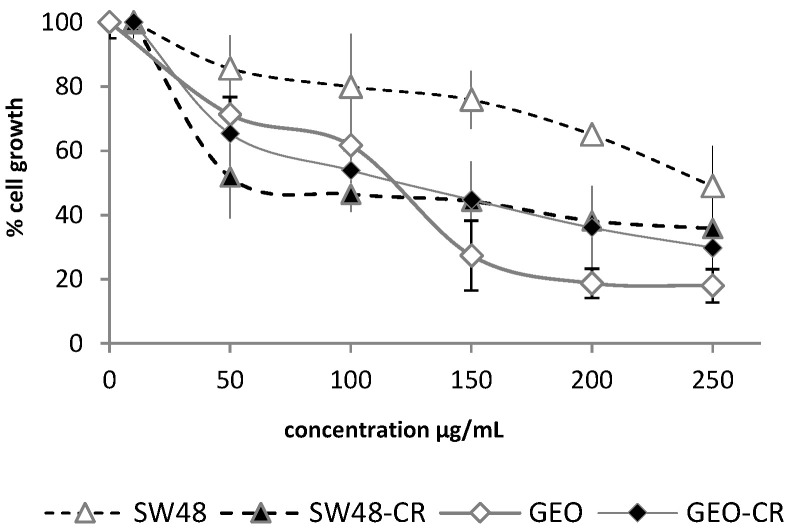
Antiproliferative activity evaluation of *O. diffusa* against the SW48, SW48-CR, GEO, and GEO-CR cell lines. Cell growth is expressed as percentage from control, and it is plotted on the vertical axis, while doses of plant extracts are reported on the horizontal axis.

**Figure 3 molecules-26-03266-f003:**
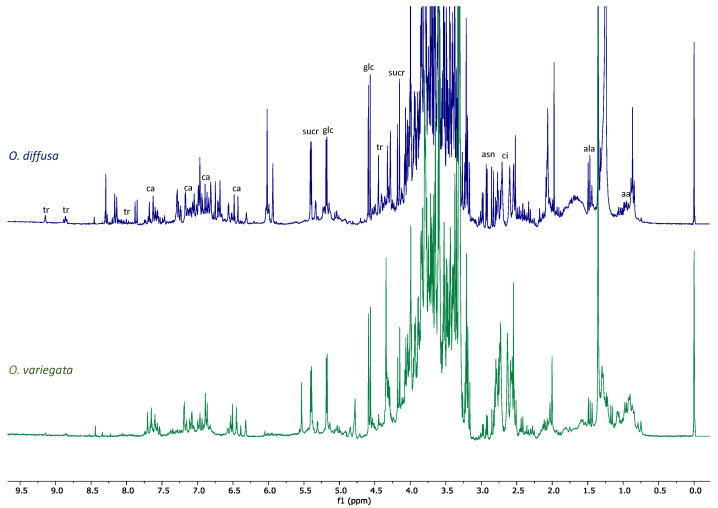
Stacked ^1^H-NMR spectra of *O. variegata* (green) and *O. diffusa* (blue). Spectra were acquired at 300 MHz, in 1:1 methanol-*d_4_*: buffer. Diagnostic signals of the main metabolites detected in both extracts are indicated on *O. variegata* spectrum by the following abbreviations: aa = amino acids, ala = alanine, asn = asparagine, ca = caffeic acid, ci = citric acid, glc = glucose, sucr = sucrose, tr = trigonelline.

**Figure 4 molecules-26-03266-f004:**
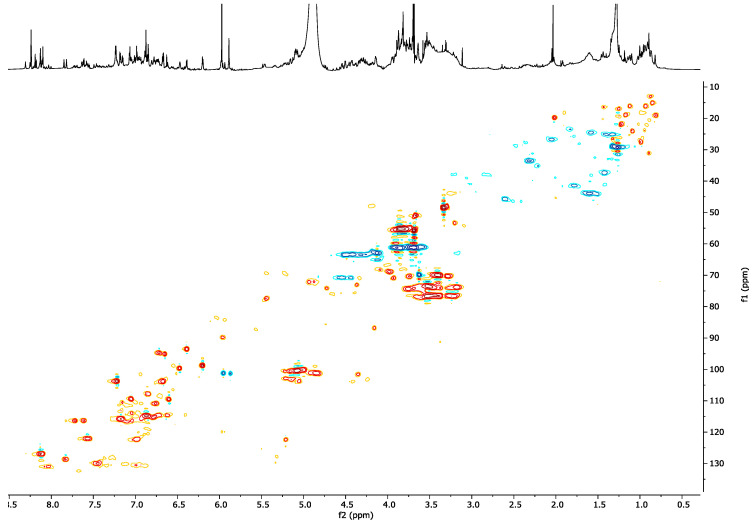
HSQC and ^1^H-NMR spectra of the OdM fraction.

**Figure 5 molecules-26-03266-f005:**
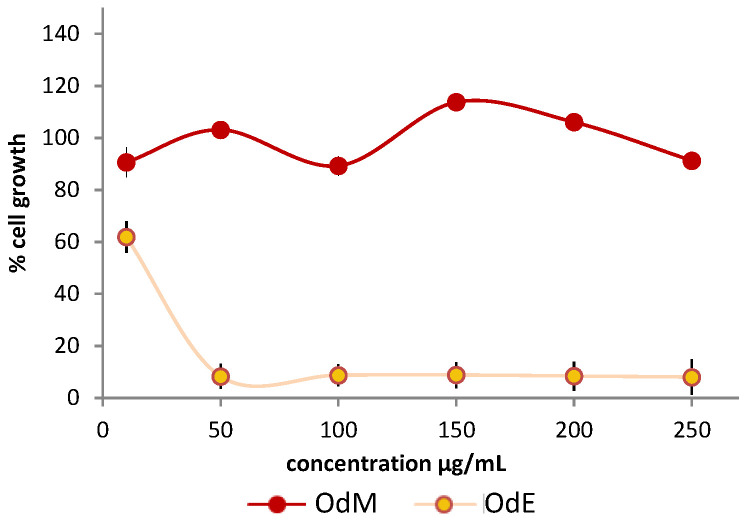
Antiproliferative activity evaluation of the *O. diffusa* partially purified fractions on HT-29 cell lines. Cell growth is expressed as percentage from control, and it is plotted on the vertical axis, while doses of plant extracts are reported on the horizontal axis.

**Figure 6 molecules-26-03266-f006:**
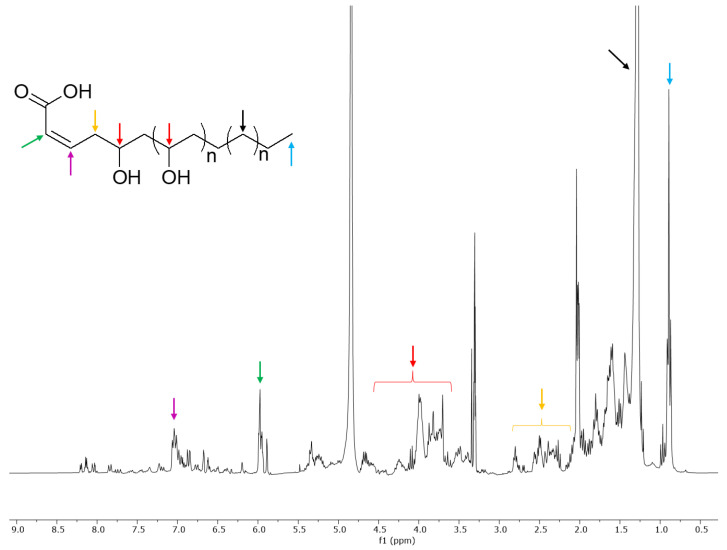
^1^H-NMR spectrum of the OdE fraction. A tentative general structure for the oxylipins is proposed, and diagnostic signals are indicated by colored arrows.

## Data Availability

The raw data supporting the conclusions of this article will be made available by the authors upon request, without undue reservation.

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
