# Peer review of "NMR Profiling of Ononis diffusa Identifies Cytotoxic Compounds against Cetuximab-Resistant Colon Cancer Cell Lines"

_molecules, 2021, doi:10.3390/molecules26113266_

Round 1

Reviewer 1 Report

The article is interesting and presents important results of cytotoxic activity of O. diffusa and O. variegata extracts. However, although it mentions a possible active metabolite responsible for the activity, they do not do any control assays with oxylipins. Nor do they study the possible mechanism of action: Oxidative stress, membrane rupture, antimetabolite or DNA fragmentation.

In the cytotoxicity assay it is also important to include "normal" cells to show selectivity against tumor lines. Its potential clinical use depends on the safety of the compounds. Low toxicity to healthy cells and high toxicity to cancer cells.

Author Response

Thank you for your comments. We are aware of the issues here raised when talking about bioactive compounds and the need for extensive studies to address all the related matters. However, as we explain in the manuscript, this work does not claim to have found an anticancer compound, but a source of potential bioactive compounds. Further studies aimed at isolating the bioactive compound(s) are currently going on and the suggested tests will be carried out on the isolated oxylipins.

Reviewer 2 Report

Dear Authors,

This work described the crude extract of a plant Ononis diffusa show cytotoxicity against cetuximab-resistant HT-29 cell lines. Furthermore, inspection of NMR spectra of crude extract/fraction hypothesize the presence of oxylipins, which leading to a conclusion that this molecule could be the potential candidate responsible for this cytotoxic activity.

1) Page 4, lines 141-142, "The aromatic and olefinic region of O. diffusa 1H-NMR spectrum showed several peaks that were not detected in O. diffusa", I believed that there is a mistake in this statement, do Authors mean O. variegata?

2) Page 4, lines 124-126, "It was clear, indeed, from the comparison of the 1H-NMR spectra (Fig. 3), that there were metabolites potentially present in both species, which would then be considered not responsible for the activity observed for O. diffusa extract.", I think this statement cannot be made casually, as there was no experimental data to support this claim. The common metabolites that present in both crude extracts in different concentrations could still be considered as the active component.

3) The lengthy discussion about the presence of oxylipins is really skeptical. After all, it is impossible to correctly assign oxylipins or any other compounds from 1H NMR spectrum in Fig. 6. 

Natural products purification is really time consuming, therefore getting a good and clean NMR spectrum is may be difficult. But to really support the claim that oxylipins or any other compounds present in crude extract, LCMS/MS profiling could further support that claim.

4) Page 6, lines 217-228, OdM fraction showed no activity, while OdE fraction was active. The OdE/ethyl acetate fraction is usually dark green color due to the present of pigments, and since this is a lipid fraction, it will contain lot of fatty acids. Therefore, these components could be the ones that show cytotoxic against HT-29 cells.

Author Response

1) Page 4, lines 141-142, "The aromatic and olefinic region of O. diffusa 1H-NMR spectrum showed several peaks that were not detected in O. diffusa", I believed that there is a mistake in this statement, do Authors mean O. variegata?

R: Thank you for pointing this out. It was corrected.

2) Page 4, lines 124-126, "It was clear, indeed, from the comparison of the 1H-NMR spectra (Fig. 3), that there were metabolites potentially present in both species, which would then be considered not responsible for the activity observed for O. diffusa extract.", I think this statement cannot be made casually, as there was no experimental data to support this claim. The common metabolites that present in both crude extracts in different concentrations could still be considered as the active component.

R: We agree, in principle, with this point. However, in this specific case, the statement was not made casually. The metabolites that are in common in the two extracts, are even the main components of the inactive extract. This further corroborates the assumption. The sentence has been slightly changed to stress this point. Furthermore, a sentence has been now added to the discussion (“These compounds (caffeic acid, caffeoyl derivatives, trigonelline and several primary metabolites) were, indeed, the main components of the O. variegata extract (Fig. 3), which however showed no activity even at the highest tested concentration”)

3) The lengthy discussion about the presence of oxylipins is really skeptical. After all, it is impossible to correctly assign oxylipins or any other compounds from 1H NMR spectrum in Fig. 6. 

Natural products purification is really time consuming, therefore getting a good and clean NMR spectrum is may be difficult. But to really support the claim that oxylipins or any other compounds present in crude extract, LCMS/MS profiling could further support that claim.

R: The presence of oxylipins is suggested not only by the signals in the 1H NMR spectrum but also (and especially) based on the 2D NMR studies on the crude extract (as reported in the results section). The spectra are also available as supplementary material.

The 2D NMR data suggest the presence of compounds that can be classified as oxylipins (therefore, we are talking about a wide range of structurally related compounds).

Identifying each single molecule in the oxylipin mixture is beyond the scope of the present work and was never claimed in the manuscript. The discussion of the NMR data even in the crude extract is, on the other hand, needed to support our conclusion. The LC-MS profile would surely help us in identifying the single molecules, but this would not add much to the present work. On the other hand, because of the very interesting biological activity, we think that isolating the pure compounds, although time-consuming, is now crucial in order to study their biological properties in detail, taking into account also all of the aspects listed by reviewer 1.

4) Page 6, lines 217-228, OdM fraction showed no activity, while OdE fraction was active. The OdE/ethyl acetate fraction is usually dark green color due to the present of pigments, and since this is a lipid fraction, it will contain lot of fatty acids. Therefore, these components could be the ones that show cytotoxic against HT-29 cells.

R: The OdE fraction derives from the fractionation of a 1:1 methanol: water extract. The hydroalcoholic mixture does not usually extract large amounts of pigments and fatty acids. Although traces of lipids and pigments might be present, they are in very low amount (and below the limit of detection in our case, otherwise we would be able to see the signals in the NMR spectra). Of course, we cannot exclude it completely that these could have a role in the observed activity even if present in such low amounts, but this will be further explored in the future studies. This is now also more clearly stated in the discussion.

Reviewer 3 Report

The study of Graziani et al is interesting, identified cytotoxic compounds from the plant Ononis diffusa against cetuximab-resistant colon cancer cell lines. Partial purification led to the identification of a fraction enriched in oxylipins, which inhibited the growth of HT-29 cell line at the concentration of 50 µg/ml.

  1. The authors have written that nearly 100% growth inhibition on HT-29 cell line. This is confusing. Instead, write the exact % growth inhibition.
  2. The authors found that the lowest concentration 50 µg/ml showed the maximum growth inhibition on HT-29 cell growth. Why the authors have not tested lower concentrations such as 5, 10 and 25 µg/ml?
  3. What about the growth inhibition of OdE fraction on other cetuximab-resistant colon cancer cell lines other than HT-29? Have the authors studied the growth inhibition of OdE fraction only on HT-29 cell line?
  4. Any in-vivo pre-clinical studies have been carried out to support the findings of the in-vitro cell growth inhibition?
  5. What kind of statistical analyses have been performed? The authors have not mentioned the statistical analyses.

Author Response

1. The authors have written that nearly 100% growth inhibition on HT-29 cell line. This is confusing. Instead, write the exact % growth inhibition.

R: thank you for the suggestion. This has been changed.

2. The authors found that the lowest concentration 50 µg/ml showed the maximum growth inhibition on HT-29 cell growth. Why the authors have not tested lower concentrations such as 5, 10 and 25 µg/ml?

R: the lowest tested concentration was 10 µg/ml as reported in the experimental section.

3. What about the growth inhibition of OdE fraction on other cetuximab-resistant colon cancer cell lines other than HT-29? Have the authors studied the growth inhibition of OdE fraction only on HT-29 cell line?

R: The partially purified fractions were only tested on the HT-29 cells for two main reasons: 1. this cell line is the one harbouring the mutation characterized by the worst outcome in mCRC patients- therefore compounds acting on this cell line are very promising; 2. This was the cell line on which the crude extracts showed the strongest activity.

4. Any in-vivo pre-clinical studies have been carried out to support the findings of the in-vitro cell growth inhibition?

R: These studies will be carried out only once the pure bioactive compounds will be isolated, and a complete structural elucidation carried out.

  1. What kind of statistical analyses have been performed? The authors have not mentioned the statistical analyses.

R: thank you for pointing this out. Treatments are different from control according to t-test. The section dedicated to statistics has been now added to the manuscript.

Round 2

Reviewer 2 Report

Dear Authors,

The major concerns have somewhat been addressed. There is some minor stuffs that need to pay attention.

1) Throughout the manuscript, (dd, J= 8.2; 2.0 Hz) should be changed to (dd, J = 8.2; 2.0 Hz), where J should be italicized.

2) The proliferation assay's method should be cited, below are some MDPI references (Marine Drugs) for MTT assay and cell cultures.

a) Mar. Drugs 201816(4), 99; https://doi.org/10.3390/md16040099

b) Mar. Drugs 201917(9), 536; https://doi.org/10.3390/md17090536

Below reference introduce a computational tool to facilitate identification of known compounds or unknown compounds that possessed similar skeleton to known compounds. This tool use HSQC spectrum for identification, for this purpose, users usually use crude extract's HSQC to explore the chemical constituents. Authors might consider using this tool for future study, by identify the known compounds, the HPLC condition for purification can be predicted.

Small Molecule Accurate Recognition Technology (SMART) to Enhance Natural Products Research. Scientific Reports volume 7, Article number: 14243 (2017).

Author Response

1) Throughout the manuscript, (dd, J= 8.2; 2.0 Hz) should be changed to (dd, J = 8.2; 2.0 Hz), where J should be italicized.

Thank you for pointing it out. It is now corrected.

2) The proliferation assay's method should be cited, below are some MDPI references (Marine Drugs) for MTT assay and cell cultures.

  1. a) Drugs201816(4), 99; https://doi.org/10.3390/md16040099
  2. b) Drugs201917(9), 536; https://doi.org/10.3390/md17090536

A reference to the method is now cited

Below reference introduce a computational tool to facilitate identification of known compounds or unknown compounds that possessed similar skeleton to known compounds. This tool use HSQC spectrum for identification, for this purpose, users usually use crude extract's HSQC to explore the chemical constituents. Authors might consider using this tool for future study, by identify the known compounds, the HPLC condition for purification can be predicted.

Small Molecule Accurate Recognition Technology (SMART) to Enhance Natural Products Research. Scientific Reports volume 7, Article number: 14243 (2017).

We highly appreciate this suggestion! Thank you!